# Effectiveness of Interventions to Manage Difficulties with Breastfeeding for Mothers of Infants under Six Months with Growth Faltering: A Systematic Review Update

**DOI:** 10.3390/nu15040988

**Published:** 2023-02-16

**Authors:** Saranya Mohandas, Ritu Rana, Barkha Sirwani, Richard Kirubakaran, Shuby Puthussery

**Affiliations:** 1Department of Public Health Programmes, Indian Institute of Public Health Gandhinagar, Gandhinagar 382042, Gujarat, India; 2Prof BV Moses Centre for Evidence Informed Health Care, Christian Medical College, Vellore 632004, Tamil Nadu, India; 3Maternal and Child Health Research Centre, Institute for Health Research, University of Bedfordshire, Luton LU1 3JU, Bedfordshire, UK

**Keywords:** infant, growth faltering, malnutrition, breastfeeding

## Abstract

(1) Background: The current evidence on management of infants under six months (u6m) with growth faltering is limited and of low quality. This review aimed at updating an existing review to inform the WHO guideline update on prevention and management of growth faltering in infants u6m. The objective is to synthesise evidence on interventions to manage breastfeeding difficulties in mothers or caregivers of infants u6m with growth faltering to improve breastfeeding practices and breastmilk intake. (2) Methods: We searched PubMed, CINAHL, and Cochrane Library from December 2018 to December 2021 for experimental studies. Using RoB 2.0 and ROBINS-I tools, we assessed study quality and results were synthesised narratively. Using the GRADE approach, we assessed the quality of evidence for four outcome domains—breastfeeding (critical), anthropometric (important), mortality (important), and morbidity (important). (3) Results: We identified seven studies, conducted among neonates (mainly preterm, n = 14 to 607), and assessed the following interventions: (a) non-nutritive sucking (NNS) on breast (n = 2) and (b) alternative supplemental feeding techniques (n = 5, cup feeding, spoon feeding, supplemental feeding tube device, and syringe feeding), and reported breastfeeding and anthropometric outcomes. None of the studies reported mortality and morbidity outcomes. The reported breastfeeding outcomes included LATCH (Latch, Audible swallowing, Type of nipple, Comfort, Hold) total score, PIBBS (Preterm Infants Breastfeeding Behaviour Scale) total score, EBF (exclusive breastfeeding) at various time points and time to transition to full breastfeeding, and reported anthropometric outcomes included weight gain and weight at different time points. Studies had ‘serious’ indirectness and ‘serious’ to ‘very serious’ risks of bias. From the limited studies we found, NNS on breast compared to NNS on finger may have some benefits on PIBBS total score; NNS on breast compared to NNS on pacifier may have some benefits on EBF at discharge; and cup feeding compared to bottle feeding may have some benefits on EBF at discharge, at three months and at six months. (4) Conclusions: Evidence on the effectiveness of interventions to manage breastfeeding difficulties in mothers or caregivers of infants u6m with growth faltering to improve breastfeeding practices and increase breastmilk intake is ‘limited’ and of ‘low’ to ‘very low’ quality. As the majority of the infants in the included studies were neonates, no new recommendations can be made for infants from one to six months due to lack of evidence in this population. We need more studies targeting infants from one to six months of age. The review was registered with PROSPERO (CRD42022309001).

## 1. Introduction

Early life malnutrition and growth faltering is a global public health problem [1]. A combination of factors—low birth weight (LBW) due to intra-uterine growth failure and/or preterm birth or anthropometric deficits (low weight-for-age, weight-for-length, or length-for-age), can lead to early life growth faltering [2,3]. Globally, each year, an estimated 14.6% births are LBW and 14.8 million babies are born preterm [4,5]. These babies have a higher risk of mortality, and if they survive, face higher risk of being wasted during early infancy—approximately 8.5 million infants aged under six months (u6m) are wasted [6].

The first six months of life represents a period of rapid maturation and development with unique dietary needs; since infants should ideally be breastfed during this period [7], the mother or carer plays a critical role in fulfilling the nutritional requirements [8]. Unmet nutritional requirements can have serious implications for infant growth and survival. The short-term implications include a higher risk of morbidity and mortality, while long-term effects have implications for health and well-being later in life such as increased risk of non-communicable diseases [7,9].

For infants with early growth faltering, the neonatal period is critical for establishing breastfeeding, but there could be challenges experienced by mothers as well as infants. Often these newborns are separated from their mothers as they are being stabilised in the neonatal intensive care units (NICUs), and this could limit opportunities for mothers to breastfeed their infants [10]. The immature and uncoordinated suckle-swallow-breathe mechanism could pose difficulties in oral feeding for infants with growth faltering [11]. Once transition to full oral feeding is achieved, infants are discharged from the NICUs, and mothers are encouraged to practice exclusive breastfeeding (EBF) at home. However, mothers require support to continue breastfeeding post-discharge, where family and societal factors play an important role in ensuring mothers continue to exclusively breastfeed these infants up to six months of age [10]. Often these mothers are compelled to start early complementary feeding around 4–5 months when they continue to face difficulties with breastfeeding [12].

To improve growth among infants with growth faltering, many early-life interventions addressing breastfeeding difficulties have been tested, ranging from cup feeding to spoon-feeding to other alternative supplemental feeding techniques [13,14,15]. In 2011 and 2013, the World Health Organization (WHO) published recommendations on the feeding of LBW infants and management of severe acute malnutrition among infants u6m, respectively; these recommendations were based on a limited and low or very low quality of evidence [16,17]. Given the importance of early infant feeding practices on morbidity, mortality, and long-term health and well-being, high-quality evidence synthesis is essential to inform prevention and management of growth failure among young infants.

The current review is an update of an existing review, ‘Feeding interventions for infants with growth failure in the first six months of life: a systematic review’ published in 2020, which aimed at informing research priorities to prevent and manage growth failure among small and at-risk infants u6m [18]. This review identified observational and experimental studies published from 1990 to 2018, focusing on infants u6m and/or their mothers, reporting on interventions/treatment modalities that improve feeding difficulties. The authors narratively synthesised a wide range of interventions (such as enteral feeding, cup feeding, and formula fortification/supplementation) and outcomes (feeding practices, anthropometry, morbidity, and mortality). The included studies aimed at improving the infant’s feeding stage, growth, or assessed the safety and efficacy of the intervention in these infants and were not always aimed at improving breastfeeding practices or breastmilk intake.

Thus, through the present review, we attempt to refine the evidence to studies assessing the effectiveness of interventions to manage breastfeeding difficulties in mothers or caregivers of infants u6m with growth faltering to improve breastfeeding practices and increase breastmilk intake. Through this review update, we aim to inform the WHO guideline update for prevention and management of growth faltering among infants u6m [19]. For this, in addition to assessing the studies included in the previous review, we assessed new studies from December 2018 to December 2021 using the revised inclusion/exclusion criteria.

## 2. Materials and Methods

### 2.1. Protocol and Ethics

This systematic review protocol was registered with the International Prospective Register of Systematic Reviews (PROSPERO) registry for systematic reviews (CRD42022309001), conducted according to the Cochrane methodology, and reporting adhered to Preferred Reporting Items for Systematic Reviews and Meta-Analyses (PRISMA) guidelines [20,21]. Due to the nature of this literature review study, ethical approval was not required.

### 2.2. Eligibility Criteria

Studies and setting: We included experimental studies—randomised controlled trials (RCTs) and non-randomised controlled trials (NRCTs), conducted in low-, middle-, and high-income countries.

Population: We included studies where the target population was mothers or caregivers of infants u6m with growth faltering who were experiencing difficulties with breastmilk intake. Infants with a potential risk of experiencing difficulties were considered as those small or at nutrition-related risk, including those with LBW (<2500 g) and those with weight loss or feeding difficulties (study authors’ definition). We excluded studies with specific groups such as babies born with certain congenital conditions.

Interventions and comparison: We included studies that assessed the effect of interventions to manage difficulties with breastfeeding for infants, mothers, or both that aimed at improving breastfeeding practices and increasing breastmilk intake. We excluded studies with the following interventions: (1) medical interventions such as use of antibiotics, or similar drugs and micronutrients in addition to human/donor/formula milk fortification, and/or (2) interventions delivered to manage difficulties with breastfeeding, however, not aimed at improving breastfeeding practices and/or breastmilk intake. We included studies where the comparison group did not include any of the aforementioned interventions (or combinations) to each other. Comparison groups with standard care were included. 

Outcomes: We included studies if they reported at least one of the breastfeeding indicator outcomes (attachment/positioning or infant breast milk intake) for infants u6m, with/without anthropometric (weight-for-length z-scores, weight-for-age z-scores, mid-upper arm circumference, change in anthropometry, weight gain), mortality, and morbidity, or recovery from co-morbidity outcomes.

### 2.3. Information Source and Search Strategy

We conducted search in three databases: PubMed, CINAHL, and Cochrane Library with limits applied for human studies, RCTs, and studies published in the English language between December 2018 and December 2021. We used predefined searches (title/abstract), Medical Subject Headings (MeSH) terms, text words, and word variants for population and intervention terms to develop a comprehensive search strategy for each database. The search strategy for PubMed is presented in Appendix B.

### 2.4. Data Collection and Analysis

Selection process: We imported identified records into the EPPI-Reviewer Web software [22] followed by removal of duplicate records (first automatically and then manually). Two reviewers independently assessed records and reports at screening and eligibility stages, any disagreements were resolved through discussion, or if required, consulting a third reviewer.

### 2.5. Data Extraction and Management

Two reviewers independently extracted data using an adapted electronic data extraction form from Cochrane [23]. The following data items were extracted for each trial: authors, year of publication, country, study characteristics, population characteristics, details of interventions, and outcome measures. Any discrepancies in extracted data were resolved by discussion or by involving a third reviewer. Final extracted data were recorded in an excel spreadsheet.

### 2.6. Assessment of Risk of Bias

We assessed outcome level risk of bias in the included studies using the Revised Cochrane Risk of Bias tool for randomised trials (RoB 2) and the Risk Of Bias In Non-randomised Studies of Interventions (ROBINS-I) tool [24,25]. For RCTs, the assessment was done on the five domains that are likely to affect the results: bias arising from the randomisation process, bias due to deviations from intended interventions, bias due to missing outcome data, bias in the measurement of the outcome, and bias in the selection of the reported result. The judgements within each domain steered to an overall risk of bias for the outcome being assessed—‘low risk’, ‘some concern’, or ‘high risk’.

We assessed the risk of bias on seven domains for NRCTs. These domains are classified under three categories: pre-intervention—which involves bias due to confounding, and selection bias; at intervention—bias in classification of interventions; and post-intervention—these are similar to the last four bias domains used in RCTs. With the help of signalling questions, we judged the risk of bias within each domain. The judgements within each domain steered to an overall risk of bias for the outcome being assessed—‘low’, ‘moderate’, ‘serious’, or ‘critical’.

Two independent reviewers conducted bias assessments and a third reviewer double-checked for accuracy. When differences in the assessment of the risk of bias existed, a consensus was reached by discussion. As the reviewers were familiar with the studies, blinding to the study author, journal of publication, or results was not possible during assessment.

### 2.7. Measures of Treatment Effect

We have presented continuous outcomes as mean difference (MD) with 95% confidence interval (CI) and dichotomous outcomes as risk ratio (RR) with 95% CI.

### 2.8. Assessment of Reporting Biases

We have less than ten studies contributing to a single comparison. Therefore, the construction of a funnel plot was not done. We conducted a comprehensive search for identifying all the published and unpublished studies.

### 2.9. Data Synthesis

There was a wide range of heterogeneity (clinical and methodological) across the studies in terms of interventions, mode of delivery, outcomes reported, and outcome measurement, hence a meta-analysis was deemed inappropriate. The analyses are hence presented as narrative synthesis.

### 2.10. Certainty of Evidence and Evidence Profiles

We used the Grading of Recommendations Assessment, Development, and Evaluation (GRADE) approach to assess the quality of evidence for four outcome domains—breastfeeding (critical), anthropometric (important), mortality (important), and morbidity (important) [26]. For each outcome, we assessed the certainty of body of evidence on four domains: (1) risk of bias, (2) inconsistency, (3) indirectness, and (4) imprecision, and rated as ‘high’, ‘moderate’, ‘low’, or ‘very low’. The quality of evidence was downgraded by one, two, or three level(s) if there were ‘serious’, ‘very serious’, or ‘extremely serious’ issues in any of the domains, respectively. We constructed GRADE evidence profiles for comparisons as appropriate using RCTs and detailed explanations are added as footnotes.

## 3. Results

### 3.1. Study Flow

Search results and the process of selection are presented in Figure 1. We identified 10,004 records. After de-duplication and screening of records, 179 (+47) reports were assessed for eligibility. Of these, 175 (+44) reports were excluded (Appendix A). Finally, seven studies were included in the review; four from the new search and three from a previous review.

### 3.2. Characteristics of Included Studies

Table 1 presents the characteristics of included studies (n = 7); three studies from the previous review [27,28,29] and four studies from the updated search [30,31,32,33]. The majority of them were RCTs (n = 5), conducted in hospital settings (NICUs) in different countries—Turkey (n = 3), India (n = 2), Egypt (n = 1), and Canada (n = 1). The number of participants ranged from 14 to 607. The studies were focused on the mother–infant dyad, mainly preterm infants, with gestational age (GA) ranging from less than 32 weeks to 37 weeks and birth weight (BW) ranging from less than 1250 g to 2300 g.

The studies reported two types of interventions—non-nutritive suckling (NNS) (n = 2) and alternative supplemental feeding techniques (n = 5). NNS included, NNS on emptied breast [30,31], while alternative supplemental feeding techniques included, supplemental feeding tube [32], spoon feeding [28], cup feeding [27,29], and syringe feeding [33].

Studies reported the following outcome measures under each domain:(1)Breastfeeding outcomes: LATCH (Latch, Audible swallowing, Type of nipple, Comfort, Hold) total score; PIBBS (Preterm Infants Breastfeeding Behaviour Scale) total score; EBF at discharge, at one week, at six weeks, at three months, and at six months; breastfeeding at six months; and time to transition to full breastfeeding.(2)Anthropometric outcomes: daily weight gain, weight gain in first 7 days, weight gain from birth until transition to breastfeeding; and weight at discharge, at the 7th day, and at the 14th day.

None of the studies reported outcomes related to mortality, morbidity, or recovery from co-morbidity.

### 3.3. Risk of Bias of Included Studies

We assessed the risk of bias for a total of seventeen outcomes from five RCTs and five outcomes from two NRCTs. The overall risk of bias for RCTs ranged from ‘some concerns’ to ‘high risk’, whereas all the NRCTs had ‘serious’ risk of bias (Appendix A).

### 3.4. Synthesis of Results

#### 3.4.1. Non-Nutritive Sucking (NNS) on Breast

We identified two RCTs assessing the effect of NNS on emptied breast on breastfeeding and anthropometric outcomes [30,31]. One RCT from India compared the effect of NNS on emptied breast with NNS on finger assessing the effect on four breastfeeding and one anthropometric outcome (Table 2), while the other trial from Canada assessed the effect of NNS on breast and NNS on pacifier on one breastfeeding outcome (Table 3).

The RCT from India enroled 14 preterm infants with GA less than 32 weeks and BW less than or equal to 1250 g [31]. The infants in the intervention arm received NNS on mother’s emptied breast with the help of a nurse for 5–10 minutes, three times a day until nutritive breastfeeding began, whereas the infants in the comparison arm received NNS on finger during gavage feeds. Compared to infants who received NNS on finger, infants who received NNS on breast had a better PIBBS score at discharge, while there was no difference between the groups in EBF at six weeks, three months, and six months. There was no difference in weight at discharge between infants in both the arms. Overall, the certainty of evidence is very low (Table 2).

The RCT from Canada enroled 33 preterm infants with GA less than 34 weeks and BW approximately 1250 to 1600 g [30]. The mothers in the intervention arm provided NNS on emptied breast without nipple shield once a day prior to scheduled enteral feedings, while infants in the comparison arm received NNS on a pacifier for 15 minutes daily. Compared to infants who received NNS on a pacifier, infants who received NNS on emptied breast had better EBF rates at discharge (low certainty of evidence, Table 3).

#### 3.4.2. Alternative Supplemental Feeding Techniques

We identified three RCTs and two NRCTs that assessed the effect of alternative supplemental feeding techniques.

The RCT from Turkey evaluated the effect of supplemental feeding tube device (SFTD) and bottle feeding on two breastfeeding and one anthropometric outcome in 46 preterm infants with GA less than 34 weeks and BW less than or equal to 1250 g [32]. The infants in the intervention arm were held by the mother and fed using a supplemental nursing system, which consisted of a feeding bottle with adjustable milk flow and neck strap (attached to the mother’s neck). The feeding tube device had two probes attached to the mother’s nipple, using which the mothers fed the infants for 20 minutes until discharge. The infants in the comparison arm were held in breastfeeding position in mother’s arms and fed using a bottle for 20 minutes. The study found no difference in LATCH score at discharge, time taken to transition to full breastfeeding and daily weight gain between the two arms (very low certainty of evidence, Table 4).

The RCT from India investigated the effects of spoon feeding and nasogastric tube feeding on 79 preterm infants of GA greater than or equal to 32 weeks and BW between 1250 and 1600 g and reported one breastfeeding and three anthropometric outcomes [28]. The infants in the intervention arm were held in the mother’s or nurse’s lap and fed using a half-filled smooth edged, sterilised, metallic spoon such that the rim of the spoon was directed towards the lip and gums, resting on the lower lip with milk line touching the upper lip, enabling the infant to lap and sip. The remaining volume of milk was given through a nasogastric tube. Infants in the comparison arm received nasogastric tube feeding only. Surprisingly, the infants who received spoon feeding took more days to reach full breastfeeding when compared to infants who received nasogastric tube feeding. Additionally, the study observed no difference in weight gain from birth until transition to full breastfeeding, weight at 7th day, and weight at 14th day of life. Overall, the certainty of evidence is very low (Table 5).

Two studies (one RCT, one NRCT) compared cup feeding with bottle feeding and reported on five outcomes—four breastfeeding and one anthropometric [27,29].

The RCT from Turkey included 607 infants with GA between 32 weeks to 35 weeks [29]. The infants in the intervention arm were held in a semi-upright position (with back and neck support) by the mother or nurse and were fed using a small medicine cup such that the edge of the cup stroked the lower lip stimulating the infant to root, until discharge. The infants in the comparison arm were fed using a bottle with a teat or nipple by the nurse until discharge. Compared to infants who received bottle feeding, infants who received cup feeding had higher rates of EBF at discharge, three months, and six months. However, there was no difference in weight gain in first seven days of study between infants of both arms. Overall, the certainty of evidence is very low (Table 6).

The NRCT from Egypt enroled 60 infants with GA between 34 weeks and 37 weeks and BW between 2000 and 2300 g [27]. The infants in the intervention arm were fed by the researchers or NICU nurse using cup feedings until discharge, while the infants in the comparison arm received bottle feedings. There was no difference in EBF at one week between infants in both arms (Appendix A).

One NRCT from Turkey included 103 preterm infants with GA less than 32 weeks and BW less than or equal to 1500 g, and compared the effects of syringe feeding with bottle feeding on two breastfeeding and anthropometric outcomes each [33]. The infants in the intervention arm received syringe feedings, with the syringe placed in middle of the infant’s tongue gently stroking and touching the palate and tongue along with stroking the mouth and gums. The infants in the comparison arm were fed with a bottle with nipple. There was no difference in breastfeeding at six months, but infants in the intervention group took less time to transition to full breastfeeding. There was no difference observed between the two groups in weight at discharge and weight gain (Appendix A).

## 4. Discussion

### 4.1. Summary of Key Findings

This review aimed at informing the update of the WHO guidelines for prevention and management of growth faltering among infants u6m. We identified seven studies—two on NNS and five on alternative supplemental feeding techniques that assessed the effectiveness of interventions to manage breastfeeding difficulties in mothers or caregivers of infants u6m with growth faltering to improve breastfeeding practices and increase breastmilk intake.

NNS on breast compared to NNS on finger may increase PIBBS total score at discharge, but the evidence is uncertain (very low quality). The evidence is very uncertain about the effect of NNS on breast on EBF at six weeks, at three months, at six months, and weight at discharge (very low quality). Further, NNS on breast compared to NNS on pacifier may result in little to no difference in EBF at discharge (low quality).

The evidence is very uncertain about the effect of SFTD compared to bottle feeding on LATCH score, time to transition to full breastfeeding, and daily weight gain (very low quality). Spoon feeding given in addition to nasogastric tube feeding compared to nasogastric tube feeding alone may increase time to transition to breastfeeding, but evidence is uncertain (very low quality). The evidence is very uncertain about the effect of spoon feeding on weight at the 7th day, weight at the 14th day, and weight gain from birth to transition to full breastfeeding (very low quality). Cup feeding compared to bottle feeding may increase EBF at discharge, at three months and six months, but the evidence is very uncertain (very low quality). The evidence is very uncertain about the effect of cup feeding on weight gain.

### 4.2. Current Evidence, Implications for Guidelines, and Need for Future Studies

Most of the current evidence is based on limited studies of ‘low’ or ‘very low’ quality [13,34,35]. A review comparing the effects of supplementation with alternative supplemental feeding devices versus supplementation with bottle feeding reported supplementation with alternative supplemental feeding devices increases full breastfeeding at discharge, at three and six months post-discharge [34]. Similarly, our review identified cup feeding (alternative supplemental feeding device) compared to bottle feeding may increase EBF at discharge, at three months and six months, but the evidence is uncertain [29]. However, similar to our findings, another review comparing cup feeding with other forms of supplemental feeding (including bottle feeding) in term and preterm infants who were unable to breastfeed, concluded there was no difference in weight gain measurements [13,29]. Another review comparing the effect of NNS on a pacifier or gloved finger with no NNS on preterm infants reported no significant effect on full breastfeeding at discharge [35]. In contrast, our review observed NNS on breast compared to NNS on a pacifier or finger improved EBF at discharge and PIBBS score at discharge, respectively [30,31]. This increase in exclusivity could be due to improved milk formation in mothers and opportunities at breast for the infant to suck and latch [36].

Given that the review is aimed at informing the update of current WHO guidelines for infants u6m with growth faltering, it is important to highlight that the identified evidence had some concerns of applicability and validity, which we present in the following section.

Interestingly, the included studies reported only in-hospital preterm and/or LBW neonates of GA ranging from less than 32 weeks to around 37 weeks, leading to ‘serious’ indirectness. This difference in population of interest and those included in studies could cause difficulties in guideline development, hence affecting the applicability. Similarly, the included studies had missing information on ‘when’, ‘how much’, and ‘how well’ the interventions were delivered. Detailed description of ‘when’ and ‘how much’ can help stakeholders to understand the applicability and replicability of these interventions, while transparent information on ‘how well’ the intervention was delivered compared to planned could help guideline developers understand the fidelity of the interventions and minimise errors while interpreting study outcomes [37].

The included studies reported on at least one breastfeeding-related outcome such as EBF at various time points, PIBBS total score, LATCH score, and time to transition to full breastfeeding, while few studies reported on anthropometric outcomes such as weight gain and weight at different time points. No studies reported mortality, morbidity, or recovery from co-morbidity, which were considered important outcomes for guideline development. There is a need for more studies reporting mortality and morbidity, given that these infants are at higher risk of mortality and short-term morbidities compared to their older counterparts [38]. The included studies used a large number of outcomes, but only very few were common. There was considerable variability in ‘how’ the outcomes were defined, measured, and reported and on ‘when’ these outcomes were measured and reported, leading to difficulties in synthesis. Furthermore, self-reported breastfeeding outcomes measured post-discharge through questionnaire or interviews may be influenced by societal expectations of breastfeeding and mothers not wanting to disappoint the care providers [13]. Future trials can address the heterogeneity in outcomes through use of standardised outcome measures, which also reduces selective reporting as all trials would define, measure, and report outcomes uniformly [39]. Furthermore, future trials should use pre-existing tools to measure outcomes that require judgements, to reduce subjectivity in outcome measurement and associated biases. Consensus on use of standardised, well-defined outcomes which are important to stakeholders, aids guideline developers in informed decision making [39].

Studies with limitations in methodology and reporting may be rated down to prevent risk of misleading results [40]. Overall, the studies had risks of bias ranging from ‘some concerns’ to ‘high risk’ across outcomes. Studies with ‘some concerns’ of bias arising from randomisation had no information on allocation concealment and unclear details to judge if the allocation sequence was random [30,31,32]. Due to nature of intervention, the participants and care providers were aware of assigned intervention, which led to ‘some concerns’ to ‘high risk’ of bias due to deviations from intended interventions in few studies [28,29,31]. The majority of outcomes in the studies had issues due to small to large missing outcome data leading to ‘high risk’ of bias [28,29,31]. Concerns of bias in measurement of outcome was predominant in breastfeeding outcomes as they were mostly observer/participant reported outcomes requiring some judgement [28,29,32]. Lack of clear information on outcome domains, measures, and analyses along with unavailability of protocol for most studies led to ‘some concerns’ of bias in selection of the reported results.

Lastly, most studies were from upper middle- to high-income countries, even though prevalence of infants u6m with growth faltering is higher in low- and lower middle-income countries.

Current (2013) guidelines recommend, “Feeding approaches for infants who are less than six months of age with severe acute malnutrition should prioritise establishing, or re-establishing exclusive breastfeeding by the mother or other caregiver (strong recommendation, very low quality evidence)” [17]. Based on the findings of our review, the guideline development group recommended inclusion of good practice statement(s) complementary to the existing guidelines. A new recommendation may be not appropriate, given the review evidence is of low to very low quality, focused on preterm, LBW infants for whom there is pre-existing guidance. In addition, the interventions in the review are not relevant to infants older than neonatal period or outside the NICU.

The current guideline (2013) as well as the update (2022–2023) will be based on evidence of low to very low quality. Hence, there is an increased need for methodologically strong trials, from low- and middle-income countries in the coming years, addressing challenges with breastfeeding, especially in infants between one and six months of age with growth faltering, so as to have a major implication on future guideline development.

### 4.3. Strengths and Limitations of the Review

The main strengths of our review are: a pre-published protocol; well-defined review question; explicit inclusion and exclusion criteria; comprehensive and updated search strategy; dual study selection, data extraction and risk of bias assessment; and rating the quality of evidence using the GRADE approach. However, there are some limitations. We used search filters for RCTs and language restriction (English) to manage the review within time limits for the guideline development group meeting. For clarity on data, efforts were made to contact authors. Furthermore, due to the heterogeneity observed in both intervention and reported outcomes, meta-analysis was not feasible.

## 5. Conclusions

Evidence on the effectiveness of interventions to manage breastfeeding difficulties in mothers or caregivers of infants u6m with growth faltering to improve breastfeeding practices and increase breastmilk intake is ‘limited’ and of ‘low’ to ‘very low’ quality. Included studies had ‘serious’ indirectness and ‘serious’ to ‘very serious’ risks of bias. As the majority of the infants in the included studies were neonates (mainly preterm), no recommendations can be made for infants from one to six months due to lack of evidence in this population. From the limited studies on neonates (preterm), we found that cup feeding compared to bottle feeding may have some benefits on EBF at discharge, at three months and at six months; NNS on breast compared to NNS on finger may have some benefits on PIBBS total score; and NNS on breast compared to NNS on pacifier may have some benefits on EBF at discharge. Studies targeting not only neonates, but also infants from one to six months of age are needed with a large sample size and standardised (measurement definition and time-points) outcome measures.

## Figures and Tables

**Figure 1 nutrients-15-00988-f001:**
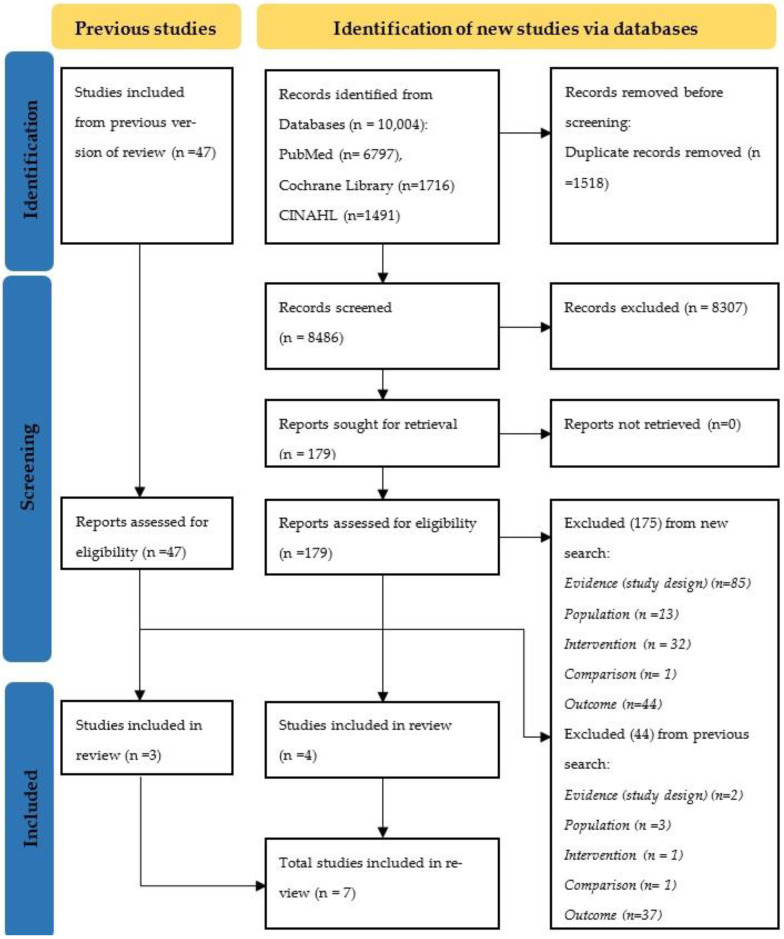
PRISMA flow diagram.

**Table 1 nutrients-15-00988-t001:** Characteristics of included studies.

Author (Year)	Study Design (Duration)	Population, Setting, Country	Intervention (Number)	Comparator (Number)	Eligible Outcomes Reported
**Non-nutritive suckling (n = 2)**					
John (2019) [31]	RCT (January–April 2015)	Mother infant pair; GA < 32 wks, BW ≤ 1250 g, NICU, India	NNS on emptied breast 5–10 m, 3 times a day + standard care (n = 7)	NNS on finger (standard care) (n = 7)	Breastfeeding, Anthropometric
Fucile (2021) [30]	RCT (Not reported)	Mother infant pair; GA < 34 wks, BW ~1250–1600 g, NICU, Canada	NNS on emptied breast once a day for 15 m (n = 17)	NNS on a pacifier (standard care) once a day for 15 m (n = 16)	Breastfeeding
**Alternative supplementary feeding techniques (n = 5)**					
Çalıkuşu (2021) [32]	RCT (August 2016–September 2017)	Mother infant pair; GA < 34 wks, BW ≤ 1250 g, NICU, Turkey	Supplemental feeding tube device, mothers breastfeed their infant for 20 m (n = 23)	Bottle feeding by mothers for 20 m (n = 23)	Breastfeeding, Anthropometric
Kumar (2010) [28]	RCT (March–October 2003)	GA ≥ 32 wks, BW 1250–1600 g, NICU, India	Spoon feeding by mother/nurse in hospital (n = 40)	Nasogastric tube feeding by nurse only (n = 39)	Breastfeeding, Anthropometric
Yilmaz (2014) [29]	RCT (April 2006–February 2008)	Mother infant pair; GA 32–35 wks BW ~1500 g, NICU, Turkey	Cup feeding using medicine cup, 15 mL of breast milk/formula (n = 308)	Bottle feeding using a bottle with a teat/nipple, breast milk/formula (n = 299)	Breastfeeding, Anthropometric
Abouelfettoh (2008) [27]	NRCT (December 2003–August 2004)	Mother infant pair; GA 34–37 wks BW ~2000–2300 g, NICU, Egypt	Cup feeding was given by researcher or nurse (n = 30)	Bottle feeding was given by researcher nurse or (n = 30)	Breastfeeding
Say (2019) [33]	NRCT (January 2013–January 2015)	GA < 32 wks, BW ≤ 1500 g, NICU, Turkey	Syringe feeding with breast milk or formula (n = 47)	Bottle feeding with nipple (n = 56)	Breastfeeding, Anthropometric

BW: birth weight; g: grams; GA: gestational age; m: minutes; ml: millilitres; NICU: neonatal intensive care unit; NNS: non-nutritive sucking; NRCT: non-randomised control trial; RCT: randomised control trial; wks: weeks.

**Table 2 nutrients-15-00988-t002:** GRADE evidence profile: Non-nutritive suckling (NNS) on breast compared to NNS on finger for infants u6m with growth faltering.

Certainty Assessment						No. of Patients		Effect		Certainty	Importance
No. of Studies	Study Design	Risk of Bias	Inconsistency	Indirectness	Imprecision	NNS Breast	NNS Finger	Relative (95% CI)	Absolute (95% CI)		
**PIBBS total score (Breastfeeding outcome)**											
1 [31]	RCT	very serious ^a^	not serious	Serious ^b^	not serious	4	5	-	MD 4 higher (0.12 higher to 7.88 higher)	⊕◯◯◯ Very low	Critical
**EBF at 6 weeks (Breastfeeding outcome)**											
1 [31]	RCT	very serious ^a^	not serious	Serious ^b^	Serious ^c^	3/4 (75.0%)	5/5 (100.0%)	RR 0.76 (0.41 to 1.42)	240 fewer per 1000 (from 590 fewer to 420 more)	⊕◯◯◯ Very low	Critical
**EBF at 3 months (Breastfeeding outcome)**											
1 [31]	RCT	very serious ^a^	not serious	Serious ^b^	Serious ^d^	2/4 (50.0%)	5/5 (100.0%)	RR 0.55 (0.22 to 1.35)	450 fewer per 1000 (from 780 fewer to 350 more)	⊕◯◯◯ Very low	Critical
**EBF at 6 months (Breastfeeding outcome)**											
1 [31]	RCT	very serious ^a^	not serious	Serious ^b^	Serious ^e^	1/4 (25.0%)	3/5 (60.0%)	RR 0.42 (0.07 to 2.63)	348 fewer per 1000 (from 558 fewer to 978 more)	⊕◯◯◯ Very low	Critical
**Weight at discharge/2 months (g) (Anthropometric outcome)**											
1 [31]	RCT	very serious ^a^	not serious	Serious ^b^	extremely serious ^f^	4	5	-	MD 25 lower (203.7 lower to 153.7 higher)	⊕◯◯◯ Very low	Important

CI: confidence interval; EBF: exclusive breastfeeding; g: grams; PIBBS: Preterm Infant Breastfeeding Behaviour Scale; MD: mean difference; RCT: randomised controlled trials; RR: risk ratio. ^a^ Downgraded by two levels for very serious risk of bias. ^b^ Downgraded by one level for serious indirectness. The target population consisted of neonates, hence not representative of u6m population. ^c^ Downgraded by one level for serious imprecision. The confidence interval ranges from 0.41 to 1.42. ^d^ Downgraded by one level for serious imprecision. The confidence interval ranges from 0.22 to 1.35. ^e^ Downgraded by one level for serious imprecision. The confidence interval ranges from 0.07 to 2.63. ^f^ Downgraded by three levels for extremely serious imprecision. The confidence interval ranges from −203.7 to 153.7. ⊕◯◯◯: very low.

**Table 3 nutrients-15-00988-t003:** GRADE evidence profile: Non-nutritive suckling (NNS) on breast compared to NNS on pacifier for infants u6m with growth faltering.

Certainty Assessment						No. of Patients		Effect		Certainty	Importance
No. of Studies	Study Design	Risk of Bias	Inconsistency	Indirectness	Imprecision	NNS Breast	NNS Pacifier	Relative (95% CI)	Absolute (95% CI)		
**EBF at discharge/1.5 months (Breastfeeding outcome)**											
1 [30]	RCT	Serious ^a^	not serious	Serious ^b^	not serious	10/16 (62.5%)	4/17 (23.5%)	RR 2.66 (1.04 to 6.78)	391 more per 1000 (from 9 more to 1000 more)	⊕⊕◯◯ Low	Critical

CI: confidence interval; RCT: randomised controlled trials; RR: risk ratio. ^a^ Downgraded by one level for serious risk of bias. ^b^ Downgraded by one level for serious indirectness. The target population consisted of neonates, hence was not representative of u6m group. ⊕⊕◯◯: low.

**Table 4 nutrients-15-00988-t004:** GRADE evidence profile: Supplemental feeding tube device (SFTD) compared to bottle feeding for infants u6m with growth faltering.

Certainty Assessment						No. of Patients		Effect		Certainty	Importance
No. of Studies	Study Design	Risk of Bias	Inconsistency	Indirectness	Imprecision	SFTD	Bottle Feeding	Relative (95% CI)	Absolute (95% CI)		
**LATCH Score (Breastfeeding outcome)**											
1 [32]	RCT	Serious ^a^	not serious	Serious ^b^	Serious ^c^	23	23	-	MD 0.13 higher (0.13 lower to 0.39 higher)	⊕◯◯◯ Very low	Critical
**Time to transition to full breastfeeding (days) (Breastfeeding outcome)**											
1 [32]	RCT	Serious ^a^	not serious	Serious ^b^	Serious ^d^	23	23	-	MD 1.3 lower (3.25 lower to 0.65 higher)	⊕◯◯◯ Very low	Critical
**Daily weight gain (g) (Anthropometric outcome)**											
1 [32]	RCT	Serious ^a^	not serious	Serious ^b^	Serious ^e^	23	23	-	MD 3.08 lower (12.6 lower to 6.44 higher)	⊕◯◯◯ Very low	Important

CI: confidence interval; g: grams; LATCH: Latch, Audible swallow, Type of nipple, Comfort, Hold; MD: mean difference; RCT: randomised controlled trial. ^a^ Downgraded by one level for serious risk of bias. ^b^ Downgraded by one level for serious indirectness. The target population consisted of neonates, hence not representative of u6m group. ^c^ Downgraded by one level for serious imprecision. The confidence intervals range from −0.13 to 0.39. ^d^ Downgraded by one level for serious imprecision. The confidence interval ranges from −3.25 to 0.65. ^e^ Downgraded by one level for serious imprecision. The confidence interval ranges from −12.60 to 6.44. ⊕◯◯◯: very low.

**Table 5 nutrients-15-00988-t005:** GRADE evidence profile: Spoon feeding compared to nasogastric feeding for infants u6m with growth faltering.

Certainty Assessment						No. of Patients		Effect		Certainty	Importance
No. of Studies	Study Design	Risk of Bias	Inconsistency	Indirectness	Imprecision	Spoon Feeding	Nasogastric Feeding	Relative (95% CI)	Absolute (95% CI)		
**Time to transition to breastfeeding (days) (Breastfeeding outcome)**											
1 [28]	RCT	very serious ^a^	not serious	Serious ^b^	not serious	40	39	-	MD 2.08 higher (0.44 higher to 3.72 higher)	⊕◯◯◯ Very low	Critical
**Weight at 7th day of life (g) (Anthropometric outcome)**											
1 [28]	RCT	Serious ^c^	not serious	Serious ^b^	very serious ^d^	39	39	-	MD 18.08 lower (68.11 lower to 31.95 higher)	⊕◯◯◯ Very low	Important
**Weight at 14th day of life (g) (Anthropometric outcome)**											
1 [28]	RCT	very serious ^e^	not serious	Serious ^b^	very serious ^f^	37	37	-	MD 3.21 higher (50.03 lower to 56.45 higher)	⊕◯◯◯ Very low	Important
**Weight gain from birth to transition to full breastfeeding (g/kg/d) (Anthropometric outcome)**											
1 [28]	RCT	Serious ^c^	not serious	Serious ^b^	Serious ^g^	40	39	-	MD 0.25 lower (2.01 lower to 1.51 higher)	⊕◯◯◯ Very low	Important

CI: confidence interval; g: grams; MD: mean difference; RCT: randomised controlled trial. ^a^ Downgraded by two levels for very serious risk of bias. There was risk of bias in outcome measurement (time to transition to breastfeeding). ^b^ Downgraded by one level for serious indirectness. The target population consisted of neonates, hence was not representative of u6m group. ^c^ Downgraded by one level for serious risk of bias. ^d^ Downgraded by two levels for very serious imprecision. The confidence interval ranges from −68.11 to 31.95. ^e^ Downgraded by two levels for very serious risk of bias. There was risk of bias due to missing outcome data. ^f^ Downgraded by two levels for very serious imprecision. The confidence interval ranges from −50.03 to 56.45. ^g^ Downgraded by one level for serious imprecision. The confidence interval ranges from −2.01 to 1.51. ⊕◯◯◯: very low.

**Table 6 nutrients-15-00988-t006:** GRADE evidence profile: Cup feeding compared to bottle feeding for infants u6m with growth faltering.

Certainty Assessment						No. of Patients		Effect		Certainty	Importance
No. of Studies	Study Design	Risk of Bias	Inconsistency	Indirectness	Imprecision	Cup Feeding	Bottle Feeding	Relative (95% CI)	Absolute (95% CI)		
**EBF at discharge (Breastfeeding outcome)**											
1 [29]	RCT	very serious ^a^	not serious	Serious ^b^	not serious	184/254 (72.4%)	123/268 (45.9%)	RR 1.58 (1.36 to 1.83)	266 more per 1000 (from 165 more to 381 more)	⊕◯◯◯ Very low	Critical
**EBF at 3 months (Breastfeeding outcome)**											
1 [29]	RCT	very serious ^a^	not serious	Serious ^b^	not serious	196/254 (77.2%)	126/268 (47.0%)	RR 1.64 (1.42 to 1.89)	301 more per 1000 (from 197 more to 418 more)	⊕◯◯◯ Very low	Critical
**EBF at 6 months (Breastfeeding outcome)**											
1 [29]	RCT	very serious ^a^	not serious	Serious ^b^	not serious	146/254 (57.5%)	113/268 (42.2%)	RR 1.36 (1.14 to 1.63)	152 more per 1000 (from 59 more to 266 more)	⊕◯◯◯ Very low	Critical
**Weight gain in first 7 days of study (g/d) (Anthropometric outcome)**											
1 [29]	RCT	very serious ^a^	not serious	Serious ^b^	Serious ^c^	254	268	-	MD 0.1 lower (0.36 lower to 0.16 higher)	⊕◯◯◯ Very low	Important

CI: confidence interval; EBF: exclusive breastfeeding; g/d: grams per day; MD: mean difference; RCT: randomised controlled trial; RR: risk ratio. ^a^ Downgraded by two levels for very serious risk of bias. ^b^ Downgraded by one level for serious indirectness. The target population consisted of neonates, hence was not representative of u6m population. ^c^ Downgraded by one level for serious imprecision. The confidence interval ranges from −0.36 to 0.16. ⊕◯◯◯: very low.

## Data Availability

Not applicable.

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
