# Peer review of "Effectiveness of Interventions to Manage Difficulties with Breastfeeding for Mothers of Infants under Six Months with Growth Faltering: A Systematic Review Update"

_nutrients, 2023, doi:10.3390/nu15040988_

Round 1
Reviewer 1 Report
Dear Authors,
Thank you for giving us the opportunity to review the manuscript. This review systematically effectiveness of interventions to manage difficulties with breastfeeding for mothers of infants under six months with growth faltering. The article is well-structured, well-written and comprehensive. After careful reading and review of “nutrients-2204749”, we propose the following review comments and suggestions.
1. It is recommended to increase the search of EMBASE database.
2. For infants with growth retardation or preterm infants, breast milk fortification is important for breastfed children, so is there a need to increase the research on breast milk fortification?
Yours
Reviewer 2 Report
The paper "Effectiveness of interventions to manage difficulties with breastfeeding for mothers of infants under six months with growth faltering: A systematic review update" is well written. However, I would not recommend authors use low-quality papers for such studies.
Minor: I think the phrase "Mainly preterm, n=14 to 607" is written by mistake. The authors need to write more clearly in the abstract and methods section how many preterm infants were analyzed in this study. If authors believe that prematurity is relevant to the study, they should mention it in the title of this paper.
